# Field-wise Learning for Multi-field Categorical Data

Zhibin Li[1], Jian Zhang[1], Yongshun Gong[1], Yazhou Yao[2], and Qiang Wu[1]

[1]University of Technology Sydney
[2]Nanjing University of Science and Technology
zhibin.li@student.uts.edu.au, jian.zhang@uts.edu.au,
yongshun.gong@student.uts.edu.au, yazhou.yao@njust.edu.cn, qiang.wu@uts.edu.au

## Abstract

We propose a new method for learning with multi-field categorical data. Multi-field categorical data are usually collected over many heterogeneous groups. These groups can reflect in the categories under a field. The existing methods try to learn a universal model that fits all data, which is challenging and inevitably results in learning a complex model. In contrast, we propose a field-wise learning method leveraging the natural structure of data to learn simple yet efficient one-to-one field-focused models with appropriate constraints. In doing this, the models can be fitted to each category and thus can better capture the underlying differences in data. We present a model that utilizes linear models with variance and low-rank constraints, to help it generalize better and reduce the number of parameters. The model is also interpretable in a field-wise manner. As the dimensionality of multi-field categorical data can be very high, the models applied to such data are mostly over-parameterized. Our theoretical analysis can potentially explain the effect of over-parametrization on the generalization of our model. It also supports the variance constraints in the learning objective. The experiment results on two large-scale datasets show the superior performance of our model, the trend of the generalization error bound, and the interpretability of learning outcomes. Our code is available at https://github.com/lzb5600/Field-wise-Learning.

## 1 Introduction

There are many machine learning tasks involving multi-field categorical data, including advertisement click prediction [1], recommender system [2] and web search [3]. An example of such data is presented in Table 1. Gender, Country, Product, and Publisher are four fields. Male and Female are two categorical features/categories under field Gender. Such data are usually converted to binary vectors through one-hot encoding and then fed into downstream machine learning models.

The scale of modern datasets has become unprecedentedly large. For multi-field categorical datasets, more samples will normally involve more categorical features. More categorical features in the dataset indicates more heterogeneity and higher dimensionality of feature vectors. To date, most effort on learning with multi-field categorical data has been put on learning a universal model [4–7], despite the fact that those data are often collected over many heterogeneous groups. Learning a universal model that fits all the heterogeneous samples will be a challenging task and usually requires complex models. To address this issue, we propose a *field-wise learning* method for multi-field categorical data, which leverages the natural structure of categorical data to form one-to-one *field-focused models*. Specifically, the models will be correlatively learned from different categories for the same field. Such an approach is called field-wise learning. In this paper, we make use of linear models as the basic field-wise models and constrain the variances and rank of corresponding weight matrices in a field-wise way. This helps promote the generalization ability and reduce the model parameters,

Table 1: An example of multi-field categorical data for advertisement click prediction.

| Clicked | Gender | Country | Product | Publisher |
|---------|--------|---------|---------|-----------|
| No | Male | Australia | Lipstick | YouTube |
| No | Male | Thailand | Snowboard | Google |
| Yes | Female | Switzerland | Lipstick | Yahoo |

which is theoretically justified. Similar to most of the models for multi-field categorical data, our model is over-parameterized. We provide a potential way to explain why over-parameterization can help the generalization of our model. In terms of interpretability, our model can provide a field-wise interpretation of the learning outcomes.

**A motivating example.** An example of multi-field categorical data is presented in Table 1. The related task is predicting whether a user will click an advertisement or not. Gender, Country, Product, and Publisher are four fields. From the Gender point of view, Male will be less likely to click the advertisement for Lipstick than Female. If we could learn two predictive models for Male and Female respectively, then we can easily capture more such biases. We define the two models as *field-focused models* for the field Gender. Similarly, people in Thailand will hardly need a snowboard compared to people in Switzerland. Leaning each Country a model can also make prediction easier compared to put all countries together. Appropriately combine all these field-focused models and we can get the final prediction. We define such a learning strategy as *field-wise learning*.

As the number of data instances grows, more countries, products, and publishers will occur. It is also common that a dataset contains many fields. This results in two practical issues for field-wise learning when building the field-focused models: 1) In some cases, the cardinality of a field can be extremely large, leading to insufficient data for learning some field-focused models; 2) the model size could become prohibitively large. To alleviate these issues, we adopt linear models with variance and low-rank constraints. Linear models require fewer data to fit and bring extra benefits to interpretability. The variance constraints are posed on the weight vectors of linear models under the same field, to ensure that the models won't deviate too much from their population mean and thus helps the learning process. Low-rank constraints can help to reduce the number of parameters and force some correlation between the linear models to facilitate the learning process.

**Alternative approaches and related work.** The sparseness and high-dimensionality of multi-field categorical data make it difficult to accurately model feature-interactions, so linear models such as Logistic regression are widely used in related applications [8]. They can be easily extended to include some higher-order hand-crafted features to improve performance. Factorization Machine (FM) [9] model has been developed to better capture the higher-order feature-interactions without a sophisticated feature engineering process. It learns the feature-interactions by parameterizing them into products of embedding vectors. FM and its neural-network variants [10, 11, 5] have become very popular in various applications involving multi-field categorical data [12–14] although they are not designed specifically for categorical data. Besides, some tree-based models [15, 16] also provide solutions for learning with multi-field categorical data. They explore very high order feature combinations in a non-parametric way, yet their exploration ability is restricted when the feature space becomes extremely high-dimensional and sparse [5]. The aforementioned models are broadly adopted universal models, while it is difficult to design and learn such models especially with the increasing heterogeneity of data.

Methods of learning multiple models have been extensively discussed within the frameworks of multi-view learning [17], linear mixed models [18], and manifold learning, while the special structure of multi-field categorical data is often overlooked. Locally Linear Factorization Machines [19, 20] combine the idea of local coordinate coding [21] and FM model to learn multiple localized FMs. The FMs could be adaptively combined according to the local coding coordinates of instances. However, their performance relies on a good distance metric. For multi-field categorical data, learning a good distance metric is as hard as learning a good predictive model. This adversely impacts the performance of such models. Other methods like sample-specific learning [22] requires auxiliary information to learn the distance metric, which limits their application when such information is not available.

To summarize, our contribution is three-fold: 1) We propose the method of field-wise learning for multi-field categorical data, which leverages the natural structure of data to learn some constraint field-focused models; 2) we design an interpretable model based on linear models with variance and low-rank constraints; 3) we prove a generalization error bound which can potentially explain why over-parameterization can help the generalization of our model.

## 2 Methodology

In this paper, we assume the dataset consists of purely categorical features. The method could be extended to handle both categorical and continuous features by transforming the continuous features to categorical features or conducting field-wise learning only on categorical fields. We formulate our method for the binary classification task, although it can also be applied to other tasks with the objective function chosen accordingly.

Given a data instance consists of $m$ categorical features from $m$ fields, we firstly convert each categorical feature to a one-hot vector through one-hot encoding. Then concatenate these one-hot vectors to form the feature vector $\mathbf{x} = [\mathbf{x}^{(1)^\top}, \mathbf{x}^{(2)^\top}, ..., \mathbf{x}^{(m)^\top}]^\top$, where $\mathbf{x}^{(i)} \in \mathbb{R}^{d_i}$ is the one-hot vector for $i$-th categorical features. The dimension $d_i$ indicates the cardinality of $i$-th field and $d = \sum_{i=1}^m d_i$ denotes the total number of features in a dataset. For example, for the first data instance in Table 1, $\mathbf{x}^{(1)^\top} = [1, 0]$ can be obtained after one-hot encoding the categorical feature "Male", and similarly for the last instance "Female" would be encoded as $\mathbf{x}^{(1)^\top} = [0, 1]$. We use $\mathbf{x}^{(-i)} = [\mathbf{x}^{(1)^\top}, ..., \mathbf{x}^{(i-1)^\top}, \mathbf{x}^{(i+1)^\top}, ..., \mathbf{x}^{(m)^\top}]^\top$ to denote the feature vector excludes $\mathbf{x}^{(i)}$.

### 2.1 Field-wise Learning

The output score of a data instance regarding $i$-th field is given by:

$$f^{(i)}(\mathbf{x}) = \mathbf{x}^{(i)^\top} g^{(i)}(\mathbf{x}^{(-i)}) \tag{1}$$

where $g^{(i)}(\mathbf{x}^{(-i)}) = [g_1^{(i)}(\mathbf{x}^{(-i)}), g_2^{(i)}(\mathbf{x}^{(-i)}), ..., g_{d_i}^{(i)}(\mathbf{x}^{(-i)})]^\top$ and $g_k^{(i)} \in g^{(i)}$ is the $k$-th component function of $g^{(i)}$. We define functions in $g^{(i)}$ as field-focused models for $i$-th field. The idea behind the formulation is clear: the method will select the associated decision function from $g^{(i)}$ according to the input $d_i$-dimensional feature vector $\mathbf{x}^{(i)}$, and apply it to the feature vector $\mathbf{x}^{(-i)}$. In doing this, we choose the decision function for the input categorical feature under $i$-th field accordingly. This enables us to apply a specialized model for each categorical feature, and thus can often simplify the component functions of $g^{(i)}$ while make them more suitable for related data instances.

Proceed with every field, we can get $m$ field-wise scores as $f(\mathbf{x}) = [f^{(1)}(\mathbf{x}), f^{(2)}(\mathbf{x}), ..., f^{(m)}(\mathbf{x})]^\top$, and the final score $\hat{y}$ will be given by:

$$\hat{y} = F(f(\mathbf{x})) \tag{2}$$

with $F$ a function to combine the field-wise decision scores in $f(\mathbf{x})$. $F$ can be a majority vote, weighted sum, or simply a sum function.

Given a loss function $\ell$, a dataset of $n$ labeled instances $\{(\mathbf{x}_j, y_j)\}_{j=1}^n$ with elements in $\mathcal{X} \times \{-1, +1\}$ and $\mathcal{X}$ a subset of $\mathbb{R}^d$, the learning objective can be formulated as:

$$\min_\theta \frac{1}{n} \sum_{j=1}^n \ell(\hat{y}_j, y_j) + \lambda \sum_{i=1}^m \text{Reg}(g^{(i)}) \tag{3}$$

where $\hat{y}_j$ is the decision score for $\mathbf{x}_j$ calculated by Eq.(1) and (2). The regularization term $\text{Reg}(g^{(i)})$ measures the complexity of $g^{(i)}$, which should be defined according to the model class of $g^{(i)}$. It can either be some soft constrains weighted by $\lambda$ as in Eq.(3) or some hard constraints on the complexity of $g^{(i)}$. Through regularizing the complexity of $g^{(i)}$, the variability of component functions in $g^{(i)}$ is controlled. As the cardinality of $i$-th field grows large, the number of component functions in $g^{(i)}$ also becomes large, while some categorical features rarely appear in training dataset because of lack of related data instances. Such constraint also enables learning of models related to these rare categorical features.

## 2.2 An implementation with linear models

In practice, we need to explicitly define the loss function $\ell$, component functions in $g^{(i)}$, the combination function $F$, and the regularization term. Here we introduce a simple yet efficient implementation of field-wise learning with linear models and give a detailed description of related variance and low-rank constraints.

Define the $k$-th component function of $g^{(i)}$ using a linear model:

$$g_k^{(i)}(\mathbf{x}^{(-i)}) = \mathbf{w}_k^{(i)\top}\mathbf{x}^{(-i)} + b_k^{(i)}, \forall k \in [1, d_i] \tag{4}$$

with $\mathbf{w}_k^{(i)} \in \mathbb{R}^{d-d_i}$ and $b_k^{(i)} \in \mathbb{R}$ denoting the weight vector and bias of the linear model respectively. Define $W^{(i)} = [\mathbf{w}_1^{(i)}, \mathbf{w}_2^{(i)}, ..., \mathbf{w}_{d_i}^{(i)}]$ and $\mathbf{b}^{(i)} = [b_1^{(i)}, b_2^{(i)}, ..., b_{d_i}^{(i)}]^\top$ so that

$$g^{(i)}(\mathbf{x}^{(-i)}) = W^{(i)\top}\mathbf{x}^{(-i)} + \mathbf{b}^{(i)}. \tag{5}$$

Choose $F$ to be a sum function so that $F(f(\mathbf{x})) = \sum_{i=1}^m f^{(i)}(\mathbf{x})$. Substitute these functions into Eq.(1) and (2) to obtain the decision score $\hat{y}$ of a data instance:

$$\hat{y} = \sum_{i=1}^m \mathbf{x}^{(i)\top}(W^{(i)\top}\mathbf{x}^{(-i)} + \mathbf{b}^{(i)}) \tag{6}$$

Adopt the Logloss defined as $\ell(\hat{y}, y) = \log(1 + \exp(-\hat{y}y))$[1] and we obtain the learning objective over $n$ labeled instances $\{(\mathbf{x}_j, y_j)\}_{j=1}^n$ as:

$$\min_{\{W^{(i)}, \mathbf{b}^{(i)}\}_{i=1}^m} \frac{1}{n}\sum_{j=1}^n \ell(\hat{y}_j, y_j) + \lambda \sum_{i=1}^m (||W_b^{(i)} - \bar{\mathbf{w}}_b^{(i)}\mathbf{1}_{d_i}^\top||_F^2 + ||\bar{\mathbf{w}}_b^{(i)}||_F^2)$$

$$\text{s.t. } W^{(i)} = U^{(i)\top}V^{(i)}, W_b^{(i)} = [W^{(i)\top}, \mathbf{b}^{(i)}]^\top \tag{7}$$

$$U^{(i)} \in \mathbb{R}^{r\times(d-d_i)}, V^{(i)} \in \mathbb{R}^{r\times d_i}, \forall i \in [1, m],$$

where $|| \cdot ||_F$, $\bar{\mathbf{w}}_b^{(i)}$, and $\mathbf{1}_{d_i}$ denotes the Frobenius norm, the column average of $W_b^{(i)}$, and a $d_i$-dimensional column vector of all ones respectively. In order to constrain the complexity of $g^{(i)}$ as the term $\text{Reg}(g^{(i)})$ in Eq.(3), we add the variance constraint on $W_b^{(i)}$ given by the term $||W_b^{(i)} - \bar{\mathbf{w}}_b^{(i)}\mathbf{1}_{d_i}^\top||_F^2$. It is related to variance of columns in $W_b^{(i)}$ so we call it a *variance constraint*. We also constrain the norm of $\bar{\mathbf{w}}_b^{(i)}$ to promote the generalization ability. These terms are weighted by $\lambda$. The *low-rank constraint* on $W^{(i)}$ helps to reduce the model parameters and reduce the computational complexity. It also ensures that the weight vectors in $W^{(i)}$ are linearly dependent to facilitate the learning. This constraint is achieved through decomposing $W^{(i)}$ into product of two rank $r$ matrices $U^{(i)}$ and $V^{(i)}$.

**Optimization.** We optimize the objective in Eq.(7) regarding $U^{(i)}$ and $V^{(i)}$ to facilitate the optimization on $W^{(i)}$ with low-rank constraint. As many modern toolkits have provided the functionality of automatic differentiation, we omit the details of derivatives here and put them in the supplementary materials. We employ stochastic gradient descent with the learning rate set by the Adagrad [23] method which is suitable for sparse data.

**Complexity analysis.** The time complexity of calculating the prediction of a data instance is given by $O(mrd+d)$. Without the variance and norm regularization, the time complexity of calculating the gradients associated with a data instance is given by $O(mrd+d)$ as well. The space complexity for our model is also an $O(mrd+d)$ term. They are both linear in the feature dimension $d$. Calculating gradients regarding the variance and norm regularization term can be as costly as $O(mr^2d+d)$, but we do not need to calculate them in every iteration, so this does not add much time complexity. The regularization term works on all weight coefficients, but the sparsity of data makes some features absent in a data mini-batch, and thus we do not need to regularize the associated weight coefficients. Therefore, we can calculate the gradients regarding regularization term every, for example, 1000 iterations so that most of the features have been presented.

# 3 Generalization error analysis

As the feature dimension $d$ can be very large for multi-field categorical data, the number of parameters of our model can be comparable to the number of training instances, which is usually considered as over-parameterization. This is a common phenomenon for many models applied to multi-field categorical data, but why this can be helpful to model generalization has not been well studied. An important question is, how would our model generalizes. In this section, we theoretically analyze the generalization error of the hypothesis class related to our implementation with linear models on the binary classification task. The presented result justifies the regularization term used in the learning objective and can potentially explain the effect of over-parametrization on our model.

**Definitions.** We firstly introduce some common settings in this section. Following Section 2, we assume the data consist of purely categorical features with $m$ fields. A labeled sample of $n$ data instances is given by $S = \{(\mathbf{x}_j, y_j)\}_{j=1}^n \in (\mathcal{X} \times \{-1, +1\})$ with $\mathcal{X}$ a subset of $\mathbb{R}^d$. The feature vectors $\mathbf{x}_j$s in $S$ are combinations of one-hot vectors as the format presented in Section 2. We assume that training samples are drawn independently and identically distributed (i.i.d.) according to some unknown distribution $\mathcal{D}$. Let the hypothesis set $\mathcal{H}$ be a family of functions mapping $\mathcal{X}$ to $\{-1, +1\}$ defined by $\mathcal{H} = \{\mathbf{x} \mapsto \sum_{i=1}^m \mathbf{x}_j^{(i)\top} (W^{(i)\top} \mathbf{x}_j^{(-i)} + \mathbf{b}^{(i)}) : \forall i \in [1, m], W^{(i)} \in \mathbb{R}^{(d-d_i) \times d_i}, \mathbf{b}^{(i)} \in \mathbb{R}^{d_i}\}$. Given the loss function $\ell$, the empirical error of a hypothesis $h \in \mathcal{H}$ over the training set $S$ is defined as $\widehat{R}_S(\ell_h) = \frac{1}{m} \sum_{i=1}^m \ell(h(\mathbf{x}_i), y_i)$. The generalization error of $h$ is defined by $R_{\mathcal{D}}(\ell_h) = \mathbb{E}_{(\mathbf{x}, y) \sim \mathcal{D}} [\ell(h(\mathbf{x}), y)]$, which is the expected loss of $h$ over the data distribution $\mathcal{D}$.

We begin by presenting a useful bound on the generalization error $R_{\mathcal{D}}(\ell_h)$, which is a slight modification to [24, Theorem 3.3]. Let $\ell$ be an $L_\ell$-Lipschitz loss function ranges in $[0, c]$. Then, for any $\delta > 0$, with probability at least $1 - \delta$ over the draw of an i.i.d. sample $S$ of size $n$, the following holds for all $h \in \mathcal{H}$:

$$R_{\mathcal{D}}(\ell_h) \leq \widehat{R}_S(\ell_h) + 2L_\ell \widehat{\mathfrak{R}}_S(\mathcal{H}) + 3c\sqrt{\frac{\log\frac{2}{\delta}}{2n}} \tag{8}$$

where $\widehat{\mathfrak{R}}_S(\mathcal{H})$ denotes the empirical Rademacher complexity of the hypothesis set $\mathcal{H}$ over the sample $S$. The proof of Eq.(8) is presented in supplementary materials. As shown in Eq.(8), the Rademacher complexity plays a crucial role in bounding the generalization error. Thus, we provide an upper bound on $\widehat{\mathfrak{R}}_S(\mathcal{H})$ in following Theorem.

**Theorem 3.1** *Let* $W_b^{(i)} = [W^{(i)\top}, \mathbf{b}^{(i)}]^\top$, *and* $\bar{\mathbf{w}}_b^{(i)} \in \mathbb{R}^{d-d_i}$ *be the column average of* $W_b^{(i)}$. $\mathbf{1}_{d_i} \in \mathbb{R}^{d_i}$ *is a* $d_i$-*dimensional column vector of all ones. Given* $||W_b^{(i)} - \bar{\mathbf{w}}_b^{(i)} \mathbf{1}_{d_i}^\top||_F \leq N_1^{(i)}$ *and* $||\bar{\mathbf{w}}_b^{(i)}||_F \leq N_2^{(i)}$ *for some constants* $N_1^{(i)}$, $N_2^{(i)}$ *and* $i = 1, 2, ..., m$, *following inequality holds for* $\widehat{\mathfrak{R}}_S(\mathcal{H})$:

$$\widehat{\mathfrak{R}}_S(\mathcal{H}) \leq \sqrt{\frac{m}{n}} \sum_{i=1}^m (N_1^{(i)} + N_2^{(i)}). \tag{9}$$

Due to space limit, the proof of Theorem 3.1 is postponed to supplementary materials. As shown in Eq.(9), bounding $\widehat{\mathfrak{R}}_S(\mathcal{H})$ relies on the bound of $||W_b^{(i)} - \bar{\mathbf{w}}_b^{(i)} \mathbf{1}_{d_i}^\top||_F$. This indicates that a small variance of the weight vectors for the field-focused models will be beneficial to the model generalization. Together with the bound on $\bar{\mathbf{w}}_b^{(i)}$, which is the column average of $W_b^{(i)}$, Eq.(9) also indicates a preference on smaller norm for each weight vector. Theorem 3.1 explains why we adopt a similar regularization term in our objective function.

**Discussion.** Substitute $\widehat{\mathfrak{R}}_S(\mathcal{H})$ with the bound in Theorem 3.1, we obtain the bound for the generalization error $R_{\mathcal{D}}(\ell_h)$ by Eq.(8) . For the assumptions made by Eq.(8), most of loss functions are $L_\ell$-Lipschitz and practically bounded in $[0, c]$. For example, the Logloss is 1-Lipschitz, and can be thought of being bounded by a positive scalar $c$ in practice.

Eq.(9) also provides a way to investigate the relationship between number of parameters and generalization error bound. The number of parameters of our model depends on the rank $r$ of $W^{(i)}$, because $W^{(i)}$ is expressed in terms of $U^{(i)}$ and $V^{(i)}$. Normally $N_1^{(i)}$ and $N_2^{(i)}$ in the bound are unknown, but

Table 2: Statistics of the datasets.

| Dataset | #instances | #fields | #features |
|---------|-----------|---------|-----------|
| Criteo | 45,840,617 | 39 | 395,894 |
| Avazu | 40,428,967 | 22 | 2,018,012 |

we can estimate them by $N_1^{(i)} \approx ||W_b^{(i)} - \bar{\mathbf{w}}_b^{(i)} \mathbf{1}_{d_i}^\top||_F$ and $N_2^{(i)} \approx ||\bar{\mathbf{w}}_b^{(i)}||_F$ from the trained model parameters $W_b^{(i)}$, as commonly done in related literature [25]. For models attain similar training loss $\widehat{R}_S(\ell_h)$ with different number of parameters, we can compare their bounds on $\widehat{\mathfrak{R}}_S(\mathcal{H})$ to compare their generalization error bounds.

# 4 Experiments

In this section, we present the experiment settings and results on two large-scale multi-field categorical datasets for advertisement click prediction, and empirically investigate the trend of generalization error bound with regard to number of parameters. We also show how to interpret our model in a field-wise manner.

## 4.1 Datasets and preprocessing

**Criteo**[2]: This dataset contains 13 numerical feature fields and 26 categorical feature fields. Numerical features were discretized and transformed into categorical features by log transformation which was proposed by the winner of Criteo Competition [1]. Features appearing less than 35 times were grouped and treated as one feature in corresponding fields. In doing this, we found the results can be slightly improved. The cardinalities of the 39 fields are 46, 97, 116, 40, 221, 108, 81, 54, 91, 9, 29, 37, 53, 1415, 552, 56354, 52647, 294, 16, 11247, 620, 4, 26104, 4880, 56697, 3154, 27, 9082, 57057, 11, 3954, 1842, 5, 56892, 16, 16, 28840, 69, 23117, respectively.

**Avazu**[3]: This dataset contains 22 categorical feature fields. Similarly, features appearing less than 4 times were grouped and treated as one feature in corresponding fields. The cardinalities of the 22 fields are 241, 8, 8, 3697, 4614, 25, 5481, 329, 31, 381763, 1611748, 6793, 6, 5, 2509, 9, 10, 432, 5, 68, 169, 61, respectively.

We summarize the statistics of these datasets in Table 2. We randomly split the data into the train (80%), validation (10%) and test (10%) sets.

## 4.2 Evaluation metrics

We adopted two commonly used metrics for advertisement click prediction: Logloss (binary cross-entropy loss) and AUC (Area Under the ROC curve). A small increase of AUC or an improvement of **0.001** on Logloss is considered to be significant in click prediction tasks [26, 27]. As a company's daily turnover can be millions of dollars, even a small lift in click-through rate brings extra millions of dollars each year.

## 4.3 Baselines, hyper-parameter settings and implementation details

We compared our model with several baselines, including the linear model, tree-based model, FM-based models, and neural networks. These models are widely used for multi-field categorical data.

- LR is the Logistic regression model with L2-regularization.
- GBDT is the Gradient Boosted Decision Tree method implemented through LightGBM [15]. It provides an efficient way that directly deals with categorical data. We chose the max depth of trees from $\{20, 30, 40, 50, 100\}$ and the number of leaves from $\{10^2, 10^3, 10^4\}$ for each dataset.

Table 3: Experiment results. The best results are bold. h: hours; m: minutes; M: million.

| Method | Avazu | | | | Criteo | | | |
|---|---|---|---|---|---|---|---|---|
| | Logloss | AUC | Time | #params | Logloss | AUC | Time | #params |
| LR | 0.3819 | 0.7763 | 1h36m | 2.02M | 0.4561 | 0.7943 | 1h22m | 0.40M |
| GBDT | 0.3817 | 0.7766 | 34m | trees=71 | 0.4453 | 0.8059 | 1h14m | trees=168 |
| FM | 0.3770 | 0.7855 | 1h8m | 201.80M | 0.4420 | 0.8082 | 3h20m | 32.07M |
| FFM | 0.3737 | 0.7914 | 7h20m | 341.04M | 0.4413 | 0.8107 | 8h46m | 60.57M |
| RaFM | 0.3741 | 0.7903 | 20h50m | 87.27M | 0.4416 | 0.8105 | 3h46m | 70.79M |
| LLFM | 0.3768 | 0.7862 | 30h45m | 532.75M | 0.4426 | 0.8095 | 27h43m | 52.25M |
| DeepFM | 0.3753 | 0.7880 | 1h52m | 62.96M | 0.4415 | 0.8104 | 3h58m | 12.66M |
| IPNN | 0.3736 | 0.7902 | 2h24m | 83.50M | 0.4411 | 0.8108 | 2h1m | 5.12M |
| OPNN | 0.3734 | 0.7906 | 3h29m | 83.87M | 0.4411 | 0.8109 | 5h49m | 5.20M |
| Ours | **0.3715** | **0.7946** | 1h31m | 357.18M | **0.4391** | **0.8129** | 2h22m | 206.65M |

- FM [9] is the original second-order Factorization Machine model. We chose the dimension for embedding vectors from $\{20, 40, 60, 80, 100\}$.

- FFM [1] is the Field-aware Factorization Machine model. This model is an improved version of FM by incorporating field information. The dimension for embedding vectors was selected from $\{2, 4, 8, 16\}$.

- RaFM [28] is the Rank-Aware Factorization Machine model. It allows the embedding dimensions of features to be adaptively adjusted according to varying frequencies of occurrences. We set the candidate set for embedding dimensions as $\{32, 64, 128\}$.

- LLFM [19] is the Locally Linear Factorization Machine model. It learns multiple localized FMs together with a local coding scheme. We chose the number of anchor points from $\{2, 3, 4, 5\}$ and dimension for embedding vectors from $\{16, 32, 64\}$.

- DeepFM [10] uses FM as a feature extractor for the connected neural networks. We use a neural network with 3 hidden layers and 400 neurons per layer as recommended in the original paper. We chose the dropout rate from $\{0.1, 0.3, 0.5, 0.7\}$. The embedding dimension for FM part was chosen from $\{10, 30, 60\}$.

- IPNN and OPNN [7] are the product-based neural networks utilize inner and outer product layer respectively. They are designed specifically for multi-field categorical data. We set the network structure as recommended in the original paper and selected the dropout rate from $\{0.1, 0.3, 0.5, 0.7\}$.

We implemented our model using PyTorch [29]. The gradients regarding the regularization term were calculated every 1000 iterations to speed up the training. The weight $\lambda$ for regularization term was selected from $\{10^{-3}, ..., 10^{-8}\}$. For setting the rank $r$ in our model, we tried two strategies: 1) chose different rank for different fields by $r_i = \log_b d_i$ and selected $b$ from $\{1.2, 1.4, 1.6, 1.8, 2, 3, 4\}$; 2) set the rank to be the same for all fields and selected $r$ from $\{4, 8, 12, ..., 28\}$. The first strategy produced different rank constraints for field-focused models under different fields, such that a field with more field-focused models would be assigned a larger rank. For both strategies, if $r_i > d_i$ we set $r_i = d_i$. We tried both strategies and selected the best based on validation sets.

For all models, the learning rates were selected from $\{0.01, 0.1\}$ and the weight decays or weights for L2-regularization were selected from $\{10^{-3}, ..., 10^{-8}\}$ where applicable. We chose all these hyper-parameters from a reasonable large grid of points and selected those led to the smallest Logloss on the validation sets. We applied the early-stopping strategy based on the validation sets for all models. All models except GBDT used the Adagrad[23] optimizer with a batch size of 2048. The optimal hyper-parameter settings of all models can be found in supplementary materials. All experiments were run on a Linux workstation with one NVIDIA Quadro RTX6000 GPU of 24GB memory.

## 4.4 Results and discussion

The experiment results are presented in Table 3. We repeated each experiment for 5 times to report the average results. The standard deviations for most models are small (about 0.0001∼0.0002) and listed

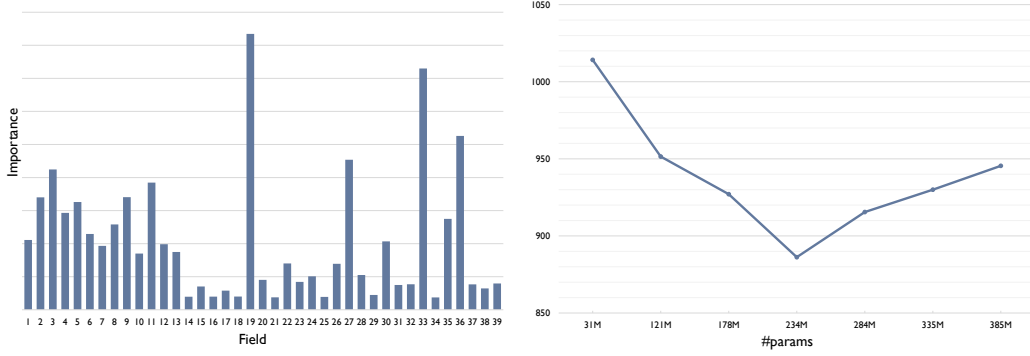

(a) Importance of each field

(b) Trend of $\sum_{i=1}^{m}(N_1^{(i)} + N_2^{(i)})$ regarding #params

Figure 1: Analysis of the models on Criteo dataset

in supplementary materials. Table 3 also shows the #params (number of parameters) and training time of each model for comparison of their complexity. For the GBDT model we list the average number of trees in the column #params.

Notice that an improvement of 0.001 on Logloss is considered significant, so our model significantly outperforms other baselines in both Avazu and Criteo datasets. Although our model features a relatively large number of parameters especially on Criteo dataset due to more fields, such over-parameterization is beneficial to our model. Other models tend to over-fit the training data when increasing the number of model parameters. LLFM also involves a large number of parameters since it learns several localized FM models. It, however, does not improve FM significantly. This is likely because of the difficulty in learning a good local coding scheme on categorical data. Other baselines try to learn a universal model for all the data, making them incapable of distinguishing the underlying differences between some groups of data and leading to inferior performance. The experiment results verify the effectiveness of our model and the field-wise learning strategy. Besides, thanks to the simple model equation and fast convergence, our model can be efficiently trained within two and three hours respectively on those two datasets.

## 4.5 Interpretation of learning outcomes

It is straightforward to explain our model in a fine-grained way. By analyzing the weights of each field-focused model, we know the importance of each feature for a specific group represented by a categorical feature. Also, our model can give a field-wise interpretation of the learning outcomes. It tells us the importance of each field for the learning task given by $||W_b^{(i)} - \bar{\mathbf{w}}_b^{(i)}\mathbf{1}_{d_i}^\top||_F/d_i$, as illustrated in Figure 1a. If this term is large for a field, then the field-focused models under this field differ much on average. This indicates categorical features in this field represent rather distinct groups for the learning task, which provides valuable information if we would like to selectively focus on certain fields, for example, to lift the quality of advertising.

## 4.6 Trend of the error bound regarding the number of parameters

Figure 1b presents the trend of the bound on $\widehat{\mathfrak{R}}_S(\mathcal{H})$ expressed by $\sum_{i=1}^{m}(N_1^{(i)} + N_2^{(i)})$ with regard to the number of parameters. We trained models of different ranks with all other hyper-parameters fixed on the training set of Criteo dataset until the Logloss went below $0.420$. We then calculated the term $\sum_{i=1}^{m}(||W_b^{(i)} - \bar{\mathbf{w}}_b^{(i)}\mathbf{1}_{d_i}^\top||_F + ||\bar{\mathbf{w}}_b^{(i)}||_F)$ as an approximation of $\sum_{i=1}^{m}(N_1^{(i)} + N_2^{(i)})$. The error bound decreases initially when #params increases. Then it increases with #params. Although these bounds are loose as most of the developed bounds for over-parameterized model [25], the trend verifies that mildly over-parameterization can be helpful to reduce the generalization error, which reflects in the initial decreasing segment of the line chart.

# 5    Conclusion and future work

In this paper, we propose a new learning method for multi-field categorical data named field-wise learning. Based on this method, we can learn one-to-one field-focused models with appropriate constraints. We implemented an interpretable model based on the linear models with variance and low-rank constraints. We have also derived a generalization error bound to theoretically support the proposed constraints and provide some explanation on the influence of over-parameterization. We achieved superior performance on two large-scale datasets compared to the state-of-the-arts. The linear models could also be extended to more complex models based on our current framework. Our future work will focus on exploring an effective way to reduce the model parameters.

# 6    Broader Impact

A broader impact discussion is not applicable as it may depend on applications.

## Acknowledgments and Disclosure of Funding

The authors greatly appreciate the financial support from the Rail Manufacturing Cooperative Research Centre (funded jointly by participating rail organizations and the Australian Federal Government's Business Cooperative Research Centres Program) through Project R3.7.2 – Big data analytics for condition based monitoring and maintenance.

## Footnotes

[1] All logarithms are base e unless specified.

[2]http://labs.criteo.com/2014/02/kaggle-display-advertising-challenge-dataset/

[3]https://www.kaggle.com/c/avazu-ctr-prediction

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
