[Supplementary Material]

# Field-wise Learning for Multi-field Categorical Data
# Supplementary Material

For simplicity, we use the notations consistently with our paper.

## 1 Derivatives

Define $s = \frac{-y}{1+\exp(y\hat{y})}$. The derivatives of Logloss on one sample $(\mathbf{x}, y)$ are then given by:

$$\frac{\partial \ell(\hat{y}, y)}{\partial U^{(i)}} = sV^{(i)}\mathbf{x}^{(i)}\mathbf{x}^{(-i)^\top}, \quad \frac{\partial \ell(\hat{y}, y)}{\partial V^{(i)}} = sU^{(i)}\mathbf{x}^{(-i)}\mathbf{x}^{(i)^\top}, \quad \frac{\partial \ell(\hat{y}, y)}{\partial \mathbf{b}^{(i)}} = s\mathbf{x}^{(i)}$$

Define $K^{(i)} = U^{(i)}U^{(i)^\top}(V^{(i)} - V_{mean}^{(i)})$, $k^{(i)} = U^{(i)}U^{(i)^\top}V_{mean}^{(i)}$, and $\mathbf{b}_{diff}^{(i)} = \mathbf{b}^{(i)} - \bar{b}^{(i)}\mathbf{1}_{d_i}$. $\bar{b}^{(i)}$ is the mean of elements in $\mathbf{b}^{(i)}$. Then for the regularization term $R_1^{(i)} = ||W_b^{(i)} - \bar{\mathbf{w}}_b^{(i)}\mathbf{1}_{d_i}^\top||_F^2$ and $R_2^{(i)} = ||\bar{\mathbf{w}}_b^{(i)}||_F^2$, corresponding derivatives are:

$$\frac{\partial R_1^{(i)}}{\partial U^{(i)}} = 2(V^{(i)} - V_{mean}^{(i)})(V^{(i)} - V_{mean}^{(i)})^\top U^{(i)},$$

$$\frac{\partial R_1^{(i)}}{\partial V^{(i)}} = 2(K^{(i)} - K_{mean}^{(i)}),$$

$$\frac{\partial R_1^{(i)}}{\partial \mathbf{b}^{(i)}} = 2(\mathbf{b}_{diff}^{(i)} - \frac{\mathbf{1}_{d_i}\mathbf{1}_{d_i}^\top}{d_i}\mathbf{b}_{diff}^{(i)}),$$

$$\frac{\partial R_2^{(i)}}{\partial U^{(i)}} = 2V_{mean}^{(i)}V_{mean}^{(i)^\top}U^{(i)}, \quad \frac{\partial R_2^{(i)}}{\partial V^{(i)}} = \frac{2}{d_i}k^{(i)}\mathbf{1}_{d_i}^\top, \quad \frac{\partial R_2^{(i)}}{\partial \mathbf{b}^{(i)}} = \frac{2\bar{b}^{(i)}\mathbf{1}_{d_i}}{d_i}.$$

The subscript "mean" denotes that associated variables are vectors calculated from the column averages of corresponding matrices, and such vectors are augmented accordingly when subtraction from matrices.

## 2 Proof of Eq.(8)

We firstly apply [1, Theorem 3.3] to a composition of loss function and our hypothesis set $\mathcal{H}$ defined as $\ell \circ \mathcal{H}$. The range of $\ell \circ \mathcal{H}$ here is in $[0, c]$. This adds a $c$ before $3\sqrt{\frac{log\frac{2}{\delta}}{2n}}$ and one can easily verify this following the same steps of proof of [1, Theorem 3.3]. Next, according to Talagrand's lemma [1, Lemma 5.7], for an $L_\ell$-Lipschitz continuous function $\ell$, we have:

$$\widehat{\Re}_S(\ell \circ \mathcal{H}) \le L_\ell\widehat{\Re}_S(\mathcal{H}) \tag{1}$$

Combine Eq.(1) with [1, Theorem 3.3] and we complete the proof.

# 3 Proof of Theorem 3.1

Define $\tilde{\mathbf{x}}_j^{(-i)} = [\tilde{\mathbf{x}}_j^{(-i)\top}, 1]^\top$ and use $< \cdot, \cdot >$ to denote inner-product. By definition of Rademacher complexity and the hypothesis set $\mathcal{H}$, we have:

$$\widehat{\mathfrak{R}}_S(\mathcal{H}) = \mathbb{E}_{\boldsymbol{\sigma}}\left[\sup_{h\in\mathcal{H}} \frac{1}{n}\sum_{j=1}^{n} \sigma_j h(\mathbf{x}_j)\right] \tag{2}$$

$$= \mathbb{E}_{\boldsymbol{\sigma}}\left[\sup_{h\in\mathcal{H}} \frac{1}{n}\sum_{j=1}^{n} \sigma_j \sum_{i=1}^{m} \mathbf{x}_j^{(i)\top}(W^{(i)\top}\mathbf{x}_j^{(-i)} + \mathbf{b}^{(i)})\right] \tag{3}$$

$$\leq \frac{1}{n}\sum_{i=1}^{m}\mathbb{E}_{\boldsymbol{\sigma}}\left[\sup_{h\in\mathcal{H}} \sum_{j=1}^{n} \sigma_j \mathbf{x}_j^{(i)\top} W_b^{(i)\top}\tilde{\mathbf{x}}_j^{(-i)}\right] \tag{4}$$

$$= \frac{1}{n}\sum_{i=1}^{m}\mathbb{E}_{\boldsymbol{\sigma}}\left[\sup_{h\in\mathcal{H}} \sum_{j=1}^{n} \langle W_b^{(i)}, \sigma_j \tilde{\mathbf{x}}_j^{(-i)}\mathbf{x}_j^{(i)\top}\rangle\right] \tag{5}$$

and see that:

$$\mathbb{E}_{\boldsymbol{\sigma}}\left[\sup_{h\in\mathcal{H}} \sum_{j=1}^{n} \langle W_b^{(i)}, \sigma_j \tilde{\mathbf{x}}_j^{(-i)}\mathbf{x}_j^{(i)\top}\rangle\right] \tag{6}$$

$$= \mathbb{E}_{\boldsymbol{\sigma}}\left[\sup_{h\in\mathcal{H}} \langle W_b^{(i)} - \bar{\mathbf{w}}_b^{(i)}\mathbf{1}_{d_i}^\top, \sum_{j=1}^{n}\sigma_j \tilde{\mathbf{x}}_j^{(-i)}\mathbf{x}_j^{(i)\top}\rangle + \langle \bar{\mathbf{w}}_b^{(i)}\mathbf{1}_{d_i}^\top, \sum_{j=1}^{n}\sigma_j \tilde{\mathbf{x}}_j^{(-i)}\mathbf{x}_j^{(i)\top}\rangle\right] \tag{7}$$

$$= \mathbb{E}_{\boldsymbol{\sigma}}\left[\sup_{h\in\mathcal{H}} \langle W_b^{(i)} - \bar{\mathbf{w}}_b^{(i)}\mathbf{1}_{d_i}^\top, \sum_{j=1}^{n}\sigma_j \tilde{\mathbf{x}}_j^{(-i)}\mathbf{x}_j^{(i)\top}\rangle + \langle \bar{\mathbf{w}}_b^{(i)}, \sum_{j=1}^{n}\sigma_j \tilde{\mathbf{x}}_j^{(-i)}\rangle\right] \tag{8}$$

$$\leq \mathbb{E}_{\boldsymbol{\sigma}}\left[\sup_{h\in\mathcal{H}} ||W_b^{(i)} - \bar{\mathbf{w}}_b^{(i)}\mathbf{1}_{d_i}^\top||_F ||\sum_{j=1}^{n}\sigma_j \tilde{\mathbf{x}}_j^{(-i)}\mathbf{x}_j^{(i)\top}||_F + ||\bar{\mathbf{w}}_b^{(i)}||_F ||\sum_{j=1}^{n}\sigma_j \tilde{\mathbf{x}}_j^{(-i)}||_F\right] \tag{9}$$

$$\leq \mathbb{E}_{\boldsymbol{\sigma}}\left[\sup_{h\in\mathcal{H}} ||W_b^{(i)} - \bar{\mathbf{w}}_b^{(i)}\mathbf{1}_{d_i}^\top||_F ||\sum_{j=1}^{n}\sigma_j \tilde{\mathbf{x}}_j^{(-i)}\mathbf{x}_j^{(i)\top}||_F\right] + \mathbb{E}_{\boldsymbol{\sigma}}\left[\sup_{h\in\mathcal{H}} ||\bar{\mathbf{w}}_b^{(i)}||_F ||\sum_{j=1}^{n}\sigma_j \tilde{\mathbf{x}}_j^{(-i)}||_F\right] \tag{10}$$

$$\leq N_1^{(i)}\mathbb{E}_{\boldsymbol{\sigma}}\left[||\sum_{j=1}^{n}\sigma_j \tilde{\mathbf{x}}_j^{(-i)}\mathbf{x}_j^{(i)\top}||_F\right] + N_2^{(i)}\mathbb{E}_{\boldsymbol{\sigma}}\left[||\sum_{j=1}^{n}\sigma_j \tilde{\mathbf{x}}_j^{(-i)}||_F\right] \tag{11}$$

Notice that following inequalities hold:

$$\mathbb{E}_{\boldsymbol{\sigma}}\left[||\sum_{j=1}^{n}\sigma_j \tilde{\mathbf{x}}_j^{(-i)}\mathbf{x}_j^{(i)\top}||_F\right] \tag{12}$$

$$\leq \mathbb{E}_{\boldsymbol{\sigma}}\left[||\sum_{j=1}^{n}\sigma_j \tilde{\mathbf{x}}_j^{(-i)}\mathbf{x}_j^{(i)\top}||_F^2\right]^{\frac{1}{2}} \tag{13}$$

$$= \left(\sum_{j=1}^{n}||\tilde{\mathbf{x}}_j^{(-i)}\mathbf{x}_j^{(i)\top}||_F^2\right)^{\frac{1}{2}} \tag{14}$$

$$= (mn)^{\frac{1}{2}} \tag{15}$$

Table 1: The hyper-parameters for each baseline. lr: learning rate; wdcy: weight decay; ebd_dim: embedding dimension or rank; a_p: number of anchor points; l2_reg: weight for L2 regularization term; dr: dropout rate.

| Method | Avazu | Criteo |
|---|---|---|
| LR | lr: 0.1, wdcy: 1e-9 | lr: 0.1, wdcy: 1e-9 |
| GBDT | num_leaves: 1e4, max_depth: 100 | num_leaves: 1e3, max_depth: 50 |
| FM | lr: 0.1, wdcy: 1e-6, ebd_dim: 100 | lr: 0.01, wdcy: 1e-5, ebd_dim: 80 |
| FFM | lr: 0.1, wdcy: 1e-6, ebd_dim: 8 | lr: 0.1, wdcy: 1e-6, ebd_dim: 4 |
| RaFM | lr: 0.01, wdcy: 1e-6, ebd_dim: {32,64,128} | lr: 0.01, wdcy: 1e-6, ebd_dim: {32,64,128} |
| LLFM | lr: 0.0001, a_p: 4, ebd_dim: 64, l2_reg:1e-6 | lr: 0.0001, a_p: 2, ebd_dim: 64, l2_reg:1e-6 |
| DeepFM | lr: 0.1, wdcy:1e-6, ebd_dim: 30, dr: 0.7 | lr: 0.1, wdcy:1e-6, ebd_dim: 10, dr: 0.3 |
| IPNN | lr: 0.01, wdcy: 1e-6, ebd_dim: 40 | lr: 0.01, wdcy: 1e-6, ebd_dim: 10 |
| OPNN | lr: 0.01, wdcy: 1e-6, ebd_dim: 40 | lr: 0.01, wdcy: 1e-6, ebd_dim: 10 |
| Ours | lr: 0.1, wdcy: 1e-8, $\lambda$: 1e-5, ebd_dim: 8 | lr: 0.01, wdcy: 1e-6, $\lambda$: 1e-3, ebd_dim: $\log_{1.6}(d_i)$ |

Table 2: Standard deviations of the Logloss reported in our paper.

| Method | Avazu | Criteo |
|---|---|---|
| LR | $0.1 \times 10^{-4}$ | $0.1 \times 10^{-4}$ |
| GBDT | $0.0 \times 10^{-4}$ | $0.0 \times 10^{-4}$ |
| FM | $2.0 \times 10^{-4}$ | $2.3 \times 10^{-4}$ |
| FFM | $0.3 \times 10^{-4}$ | $0.3 \times 10^{-4}$ |
| RaFM | $0.0 \times 10^{-4}$ | $0.0 \times 10^{-4}$ |
| LLFM | $0.0 \times 10^{-4}$ | $0.0 \times 10^{-4}$ |
| DeepFM | $0.7 \times 10^{-4}$ | $0.8 \times 10^{-4}$ |
| IPNN | $1.2 \times 10^{-4}$ | $1.0 \times 10^{-4}$ |
| OPNN | $1.0 \times 10^{-4}$ | $1.0 \times 10^{-4}$ |
| Ours | $2.0 \times 10^{-4}$ | $0.5 \times 10^{-4}$ |

The first inequality uses Jensen's inequality, and the second equality uses the property $\mathbb{E}[\sigma_i \sigma_j] = \mathbb{E}[\sigma_i]\mathbb{E}[\sigma_j] = 0$ for $i \neq j$. The last equality uses the properties that $\mathbf{x}_j^{(i)}$ is a one-hot vector and $\tilde{\mathbf{x}}_j^{(-i)}$ has exactly $m$ 1s. Follow the same steps and we can get $\mathbb{E}_{\boldsymbol{\sigma}}\left[||\sum_{j=1}^{n} \sigma_j \tilde{\mathbf{x}}_j^{(-i)}||_F\right] \leq (mn)^{\frac{1}{2}}$.

Combine above results and we can get:

$$\widehat{\mathfrak{R}}_S(\mathcal{H}) \leq \frac{1}{n}(mn)^{\frac{1}{2}} \sum_{i=1}^{m}(N_1^{(i)} + N_2^{(i)}) = \sqrt{\frac{m}{n}} \sum_{i=1}^{m}(N_1^{(i)} + N_2^{(i)}) \tag{16}$$

so we complete the proof.

## 4   Experiment details

The hyper-parameters for each baseline are presented in Table 1. Table 2 shows the standard deviations of the Logloss reported in our paper, which are based on 5 runs.

## References

[1] Mehryar Mohri, Afshin Rostamizadeh, and Ameet Talwalkar. *Foundations of machine learning*. MIT press, 2018.