[Reviews · NeurIPS 2020]

Review 1

Summary and Contributions: The proposed method employs the natural structure of data to learn simple and efficient models. The models can be fitted to each category and can better capture the underlying differences in data.

Strengths: 1, The writing of this paper is clear and easy to understand. 2, The performance of the proposed method outperforms other methods in two important evaluation metrics.

Weaknesses: 1, Some recent related methods are missed in the experiment parts. 2, Compared with IPNN and OPNN, the parameters are much large, but the performance only raise a little. The improvement of your method is not impressive. 3,The caption of Figuer 1 iis missed. 4, I want to know more about the datasets, which is different about traditional dataset,

Correctness: yes

Clarity: yes

Relation to Prior Work: yes

Reproducibility: Yes

Additional Feedback:


Review 2

Summary and Contributions: This paper proposes the field-wise learning for dealing with multi-field categorical data. Specifically, the proposed method is based on the linear models with variance and low-rank constraints. Also, it leverages the structure of data to learn the one-to-one field-focused models. A generalization error bound to theoretically support the proposed constraints as well as some explanation on the influence of over-parameterization is proposed. Experiments are conducted on the Criteo and Avazu datasets.

Strengths: + The problem studied in this paper is interesting and practical. + The paper is clearly written.

Weaknesses: - The proposed method lacks technique contributions. The main components of field-wise learning is simple and straightforwards, even popular used in other related machine learning / data mining tasks. - The generalization error bound seems tight, which cannot bring a theoritical basis. - In experiments, the datasets used are Criteo and Avazu. More real-world datas should be considered as important test beds, e.g., iPinYou, etc. - The compared methods are out-of-time. The very recent one was published in 2017. It cannot validate the effectiveness of the proposed method. Recent state-of-the-arts published in KDD, NeurIPS should be compared and discussed.

Correctness: The method and claims are good.

Clarity: This paper is easy to follow and clearly written.

Relation to Prior Work: Lacks discussions with recent state-of-the-arts as well as empirical comparisons. The authors are encouraged to discuss with them.

Reproducibility: Yes

Additional Feedback:


Review 3

Summary and Contributions: This article presents a new method for multi-field learning, i.e. learning with nominal variables. The proposed method learns a model for every field, and every value of a specific field. A low-rank regularization mechanism is applied such that the models for different values of the same field look similar. The models of different fields are aggregated to obtain the final prediction. Linear models are used for the different fields and all models are jointly optimized in a single optimization problem. The authors present generalization bounds and experimental results on two datasets.

Strengths: Overall this is an interesting method that is substantially novel. The paper is also well written. I would recommend to accept this paper, but I do have some suggestions for improvement.

Weaknesses: - some more in-depth discussion of differences with existing methods would have been useful (see below) - not very clear why the method outperforms other methods

Correctness: It is nice that the proposed method outperforms existing methods, but it is not very clear to me what the main reason is. Does the improvement come from the specific model structure, which regularizes individual models for specific values of a specific field, or does the improvement come from the ensembling effect created by the aggregation function F? If the latter is the case, then it is quite normal that the new method outperforms for example logistic regression, which fits a single model to the data.

Clarity: Yes, very well written.

Relation to Prior Work: One could argue that the method is an extension of multi-task learning methods that adopt low-rank regularization. Multi-task learning methods are often applied to one important specific field, e.g. location. The method of the authors extends this idea to multiple fields. I would have liked to read a bit more about that in this paper. Similarly, there is also a connection with factor models and mixed linear models in statistics, which are developed specifically for multi-field data. From that perspective, I found the notion multi-field data a bit confusing. These are simply called factors in statistics, so why invent a new name for something that has been thoroughly discussed in the statistics literature?

Reproducibility: Yes

Additional Feedback: In optimization problem (7) the models for the different fields all have the same rank r. Is that not a serious limitation? I can imagine that some fields only take a few values (e.g. gender) whereas others have a lot of values (e.g. location). In such situations one should use different values of r. The proposed method also only considers categorical features. Most datasets, however, have a mix of categorical and continuous features. How could the method be extended to handle both categorical and continuous features? Related to this last remark, is building a model for every individual field really the way to go when the number of fields is large?


Review 4

Summary and Contributions: The authors present an approach for modelling categorical variables. Each categorical column in a table is termed ‘field’ by the authors. The main idea appears to be based on splitting the regularisation term for each ‘field’. The authors present a thorough derivation of their method. A linear and a nonlinear model are developed. These contributions are combined with a strong experimental section that shows the strength of the proposed approach in comparison with other approaches. I enjoyed reading this paper and think it could be a relevant contribution to the field. Admittedly, when first reading it, I thought the methodological contributions are not overwhelming: there exist many similar approaches inspired by Canonical Correlation Analysis / Multi-view learning / Collective Matrix Factorization / Dictionary Learning / Imputation, see e.g. [1,2,3]. Most of these papers treat (or at least could treat) the problem of dealing with a heterogeneous mix of continuous and different categorical variables in a similar manner - without writing a separate paper about it how they model these categorical variables. But I feel this topic is important and actually does deserve more attention rather than being glossed over en passant when writing about a novel matrix factorisation or neural network method. It’s not a mere preprocessing choice how to model categorical variables, especially when they have high cardinality and are dirty (like strings with typos). As the manuscript has a great experimental section I’m leaning towards accept here. [1] Singh et al. Relational Learning via Collective Matrix Factorization http://www.cs.cmu.edu/~ggordon/singh-gordon-kdd-factorization.pdf[2] Li et al. A Survey of Multi-View Representation Learning https://arxiv.org/pdf/1610.01206.pdf [3] Wu et al, Multi-view low-rank dictionary learning for image classification, https://www.sciencedirect.com/science/article/abs/pii/S0031320315003003

Strengths: The strength of the paper is a well thought through modelling approach for heterogeneous sets of categorical variables. Another plus is the solid experimental section. I particularly liked how the authors include the number of parameters in one of the results tables.

Weaknesses: I guess the main limitation is novelty, the approach is very similar to other forms of multi-view learning, as mentioned in the summary section.

Correctness: The approach looks sensible and the derivation appears sound. The experimental validation is solid.

Clarity: yes

Relation to Prior Work: The authors compare their work to a large number of related methods and a number of competitor methods are discussed. The connection to other multi-view methods (which are different in that they don’t model categorical variables only, but are still similar in their regularization) could be discussed.

Reproducibility: Yes

Additional Feedback: As mentioned, the relation to other similar methods could be discussed better, but I think the experimental comparisons are great as they are. And it would be interesting how the proposed method would work with more complex network architectures, as alluded to in the conclusion. For reducing the number of parameters, maybe a sparsity inducing norm rather than an L_2 norm on W would work, as in [4]? [4] Mairal, Online Dictionary Learning for Sparse Coding https://www.di.ens.fr/willow/pdfs/icml09.pdf

[Author Response · NeurIPS 2020]

We appreciate all reviewers' valuable comments, and greatly encouraged by the positive comments, e.g. the problem
studied in this paper is interesting and practical; this is an interesting method that is substantially novel; the approach
looks sensible and the derivation appears sound; the experimental validation is solid. We shall address the main concerns
point by point as follows.

**To Reviewer 1 & 2 on missing recent related methods.** We had conducted extra experiments on more recent related
methods as listed in the table below, but initially we did not include them in our submission considering that it would be
too messy to list the results of all methods, which was also not allowed due to the space limit. Thus, we chose to report
those were 1) representative that served as baselines for most of the related works 2) as interpretable as our method,
although we compared to some advanced deep models to show the effectiveness of our method, and 3) scalable to the
scale of datasets in our experiments. Of course, we do agree that our submission can be more persuasive by including
more results of the SOTA methods. We would include these results in supplementary materials of the final version.

**To Reviewer 1: Q1 - The improvement of your method is not impressive.** As mentioned with references in line 224
of our paper, an improvement of 0.001 on Logloss is significant on the two datasets Avazu and Criteo, considering
the size of datasets and business value of related tasks. Our model improves Logloss by 0.002 and produces more
interpretable results compared to the second-best method which is based on neural networks. Thus we claim the results
to be significantly better, although more #params are used. **Q2 - I want to know more about the datasets, which is**
**different about traditional dataset.** The datasets are two publicly available advertisement click prediction datasets
with anonymous features and columns. Some columns contain site_id and ad_id so the dimensionality is very large. We
would include more details in supplementary materials of the final version.

**To Reviewer 2: Q1 - The proposed method lacks technique contributions.** We admit that our idea is simple and
straightforward, but based on our knowledge such learning strategy has not been explored for multi-field categorical data.
**Q2 - The generalization error bound seems tight.** Our generalization bound is tight compared to many SOTA papers
regarding over-parametrised models. For example, related bound on Criteo dataset is tighter than initialisation-based
bounds calculated with [4] as we observed in our experiments. **Q3 - More real-world data should be considered.** We
had conducted more experiments on datasets such as MovieLens and Frappe. Our model consistently outperformed
other baselines, but given the page limitation, we could only report the results on two larger datasets Avazu and Criteo.

**To Reviewer 3:** Thanks for your suggestions and we will revise our paper accordingly. **Q1 - Does the improvement**
**come from the specific model structure or the aggregation function F.** We set the aggregation function F to a simple
sum function in our paper so that we can mainly credit the performance gain to the specific model structure. **Q2 - one**
**should use different values of r.** This is exactly what we did in the experiments. We chose different ranks for each
field in a log scale regarding the cardinalities of each field as mentioned in line 254 of our paper. **Q3 - How could the**
**method be extended to handle both categorical and continuous features?** 1) we could transform the continuous
features to categorical features by log transformation as mentioned in line 209 of our paper 2) or we could conduct
field-wise learning only on categorical fields, and directly use continuous features in each field-focused model. **Q4 - is**
**building a model for every individual field really the way to go when the number of fields is large?** We may have
to selectively choose some of the fields for field-wise learning on datasets with more fields and features.

**To Reviewer 4: Q1 - The connection to other multi-view methods.** We agree with you that in the higher-level
concept the method can be related to multiview learning. Here we focus more on the special structure of categorical
variables, which is discussed less in general multiview learning method. We would add a short discussion on the
connection to multi-view methods. **Q2 - For reducing the number of parameters, maybe a sparsity inducing norm**
**rather than an L_2 norm on W would work.** Thanks for the suggestion and we may investigate it in our future work.
We did not pose L2 norm constraint on W but on column average of W to promote the generalisation ability. To reduce
the parameters, we decomposed $W$ into two much smaller low-rank matrices $U$ and $V$.

| Method | Avazu | | | | Criteo | | | |
|---|---|---|---|---|---|---|---|---|
| | Logloss | AUC | Time | #params | Logloss | AUC | Time | #params |
| RaFM [1] | 0.3774 | 0.7862 | 20h58m | 86.53M | 0.4417 | 0.8104 | 10h22m | 29.26M |
| IFM [2] | 0.3746 | 0.7885 | 30m | 82.74M | 0.4403 | 0.8118 | 3h20m | 16.07M |
| AFN [3] | 0.3768 | 0.7857 | 3h26m | 107.09M | 0.4406 | 0.8118 | 10h14m | 16.71M |

[1] Xiaoshuang Chen, et al. "RaFM: Rank-Aware Factorization Machines." ICML 2019.

[2] Yantao Yu, et al. "An Input-aware Factorization Machine for Sparse Prediction." IJCAI 2019.

[3] Weiyu Cheng, et al. "Adaptive Factorization Network: Learning Adaptive-Order Feature Interactions." AAAI 2020.

[4] Behnam Neyshabur, et al. "The role of over-parametrization in generalization of neural networks." ICLR 2019.


[Meta-Review · NeurIPS 2020]

Two reviewers have championed this paper in the discussion phase. These two reviewers provide a high fidelity assessment of this submission and the authors have provided meaningful responses to their comments. I am recommending that the authors also take the two negative scoring reviewers comments to heart in updating their manuscript for the camera-ready.